# Tumor-mediated microbiota alteration impairs synaptic tagging/capture in the hippocampal CA1 area via IL-1β production

Ying Zhu[1,6], Yu Mei [1,6], Nimmi Baby[2], Huey Yee Teo[1], Zuhairah Binte Hanafi [1], Siti Nazihah Mohd Salleh [3], Sreedharan Sajikumar [2,4,5 ✉] & Haiyan Liu [1 ✉]

Cancer patients often experience impairments in cognitive function. However, the evidence for tumor-mediated neurological impairment and detailed mechanisms are still lacking. Gut microbiota has been demonstrated to be involved in the immune system homeostasis and brain functions. Here we find that hepatocellular carcinoma (HCC) growth alters the gut microbiota and impedes the cognitive functions. The synaptic tagging and capture (STC), an associative cellular mechanism for the formation of associative memory, is impaired in the tumor-bearing mice. STC expression is rescued after microbiota sterilization. Transplantation of microbiota from HCC tumor-bearing mice induces similar STC impairment in wide type mice. Mechanistic study reveals that HCC growth significantly elevates the serum and hippocampus IL-1β levels. IL-1β depletion in the HCC tumor-bearing mice restores the STC. Taken together, these results demonstrate that gut microbiota plays a crucial role in mediating the tumor-induced impairment of the cognitive function via upregulating IL-1β production.

[1] Immunology Translational Research Programme, Department of Microbiology of Immunology, Yong Loo Lin School of Medicine, Immunology Programme, Life Sciences Institute, National University of Singapore, Singapore 117456, Singapore. [2] Department of Physiology, National University of Singapore, Singapore 117597, Singapore. [3] Human Monoclonal Antibody Platform, Singapore Immunology Network (SIgN), Agency for Science, Technology and Research (A*STAR), Singapore 138648, Singapore. [4] Life Sciences Institute Neurobiology Programme, National University of Singapore, Singapore 117456, Singapore. [5] Healthy Longevity Translational Research Programme, Yong Loo Lin School of Medicine, National University of Singapore, Singapore 117456, Singapore. [6] These authors contributed equally: Ying Zhu, Yu Mei. ✉email: phssks@nus.edu.sg; micliuh@nus.edu.sg

Hepatocellular carcinoma (HCC) is one of the most common types of liver cancer, ranking as the third most cancer-related mortality globally[1]. Over 80% of HCC cases are caused by inflammation/injury-induced liver cirrhosis, leading to the compensatory proliferation of hepatocytes[2]. Major risk factors include viral infections, diabetes, alcohol abuse, and exposure to toxins. Recently, the role of microbiota in cancer development has been well recognized. Leaky gut and dysbiosis are reported to be critical contributors to liver cancer development in patients with liver cirrhosis[3,4]. Alterations in the gut microbiota and associated inflammation in the liver have been identified as potential factors responsible for HCC progression in liver cirrhosis patients[5]. This liver associated inflammation is proposed to be primarily reliant on the activation of the innate immune response to the pathogen-associated molecular patterns (PAMPs) from the hepatic portal circulation, leading to the increased pro-inflammatory cytokines production. In a murine Diethylnitrosamine (DEN) induced HCC model, gut leakage induces bacterial translocation that results in chronic liver inflammation through LPS/TLR4 axis[6]. The contribution of microbiota to HCC growth was blocked when sterilization of intestinal microbiota with antibiotics or genetic inhibition of TLR4. Moreover, gut microbiota has also been reported to promote obesity induced HCC progression through TLR2[7]. These findings suggested that both microbial imbalance in the intestine and the bacterial translocation from the intestine to the liver may play a vital part in hepatocarcinogenesis. It is worth noting that most studies were focused on the effects of dysbiosis on HCC development, while the influence of HCC on gut microbiota has been rarely studied.

Cancer-related cognitive impairment (CRCI) is defined as non-central nervous system (CNS) cancer patients suffer from cognitive impairment symptoms, including impairment of short-term and working memory, attention, executive functions, and processing speed[8,9]. For example, more than half of the breast cancer patients are reported to have cognitive complaints during/after receiving chemotherapy[10]. Similarly, up to 40% of advanced HCC and liver cirrhosis patients can develop hepatic encephalopathy[11]. However, it remains uncertain whether CRCI is caused by the cancer itself, treatment, or physiological factors. In recent years, accumulated evidence has suggested that intestinal and extraintestinal disorders may lead to cognitive deficits, highlighting the significance of identifying the microbiota that may affect cognitive functions[12,13]. The gut–brain axis, a complex bidirectional communication network between CNS and peripheral intestinal functions, links the brain cognitive centers with intestinal microbiota. Emerging evidence suggested that intestinal microbiota greatly influences brain functions through indirectly interacting with CNS via neuroendocrine, metabolic, and inflammatory pathways[14]. For instance, germ-free mice exhibited absence of non-spatial or working memory in response to exposure to either novel object or T-maze[15]. The composition of the microbiota was associated with cognitive function development in infants[16]. Obesity associated intestinal microbiota dysbiosis may affect anxiety, depression, episodic, and semantic memory[17–19]. Several key factors have been identified to link gut–brain axis, including insulin[20], brain-derived neurotrophic factor (BDNF)[21], short-chain fatty acids (SCFAs)[22], IL-17[23], etc. Despite this progress, the precise mechanism of how microbiota affects cognitive functions remains unclear.

In this study, we aimed to investigate the effects of HCC growth on gut microbiota composition and its association with brain cognitive functions. We found that HCC growth markedly altered gut microbiota composition, resulting in impaired expression of synaptic tagging and capture (STC) in the hippocampus. We further identified IL-1β as the major cytokine responsible for the impaired brain cognitive function. Blocking IL-1β rescued the STC in the HCC tumor-bearing mice. Thus, our study presented insightful findings in the liver/tumor-gut-brain axis and its role in inducing CRCI. Targeting microbiota or IL-1β could serve as promising therapeutic approaches for CRCI treatment.

## Results

**Synaptic tagging and capture, cellular model of associative memory is impaired in HCC tumor-bearing mice.** To explore the potential effects of tumor growth on cognitive function, we established murine HCC model by injecting tumor cells into the liver. Two weeks later, hippocampus was harvested, late long-term potentiation (Late-LTP) of synaptic transmission and synaptic tagging/capture (STC) in CA1 neurons were studied (Fig. 1a). Briefly, hippocampus from mock and tumor-bearing mice were given strong high-frequency stimulation (STET) to synaptic input S1, the LTP was recorded for 240 min (Fig. 1b). We found that both the control and tumor-bearing mice showed late-LTP until 240 min, suggesting tumor growth had no influence on late-LTP (Fig. 1c, d). Next, we evaluated the difference of STC in control and tumor-bearing mice. A strong before weak (SBW) paradigm was employed to study STC, in which Late-LTP was induced in S1 by STET 30 min before the induction of early-LTP by weak tetanisation (WTET) in S2. The results showed that in control mice, both S1 and S2 maintained a significant potentiation for 240 min (Fig. 1e). However, the hippocampus isolated from tumor-bearing mice failed to express STC (Fig. 1f). Hence, our results suggested that tumor-bearing mice had impaired STC in CA1 neurons.

**Tumor-bearing mice exhibited altered gut microbiota diversity and composition.** Recent studies have highlighted the profound impact of microbiota on cancer development and cognitive dysfunction[24]. In order to investigate the potential contribution of microbiota to the impaired STC observed in the hippocampus of tumor-bearing mice, we performed 16S rRNA sequencing to identify the microbiota composition from the intestine of control and tumor-bearing mice. The taxonomy profiles at the class level revealed that the intestine bacteria community was dominated by *Bacteroidia* (55.5% on average), followed by *Clostridia* (24.8% on average), *Bacilli* (15.4% on average), and *Actinobacteria* (2.7% on average); at the genus level, the intestine bacteria community was dominated by *Muribaculaceae* (45.5% on average), *Lactobacillus* (8.6% on average), *Alistipes* (5.7% on average), *Lachnospiraceae_NK4A136_group* (5.2% on average), followed by *f__Lachnospiraceae_Unclassified* (4.4% on average), and *Dubosiella* (4.3% on average). During the HCC progression, we observed significant changes in the relative abundance of several bacterial taxa. Specifically, the relative abundance of *Bacilli* (2.4 folds on average) and *Verrucomicrobiae* (1.7 folds on average) decreased most dramatically at class level, *[Eubacterium]_fissicatena_group* (9.24 folds on average), *Dubosiella* (4.1 folds on average) and *Lactobacillus* (2.35 folds on average) at genus level. Conversely, the relative abundance of *Bacteroides* (617.1 folds on average) was most dramatically increased at genus level (Fig. 2a).

Next, we used different index to examine the alpha diversity of the microbiota in control and tumor-bearing mice. As shown in Fig. 2b, the ACE index, Shannon index and Simpson index showed no significant difference within control and tumor-bearing mice. Interestingly, we noticed that the observed species were significantly decreased in the tumor-bearing mice, suggesting the microbiota diversity was decreased during cancer progression. We also examined the taxonomic beta diversity between the groups. A significant difference was observed by

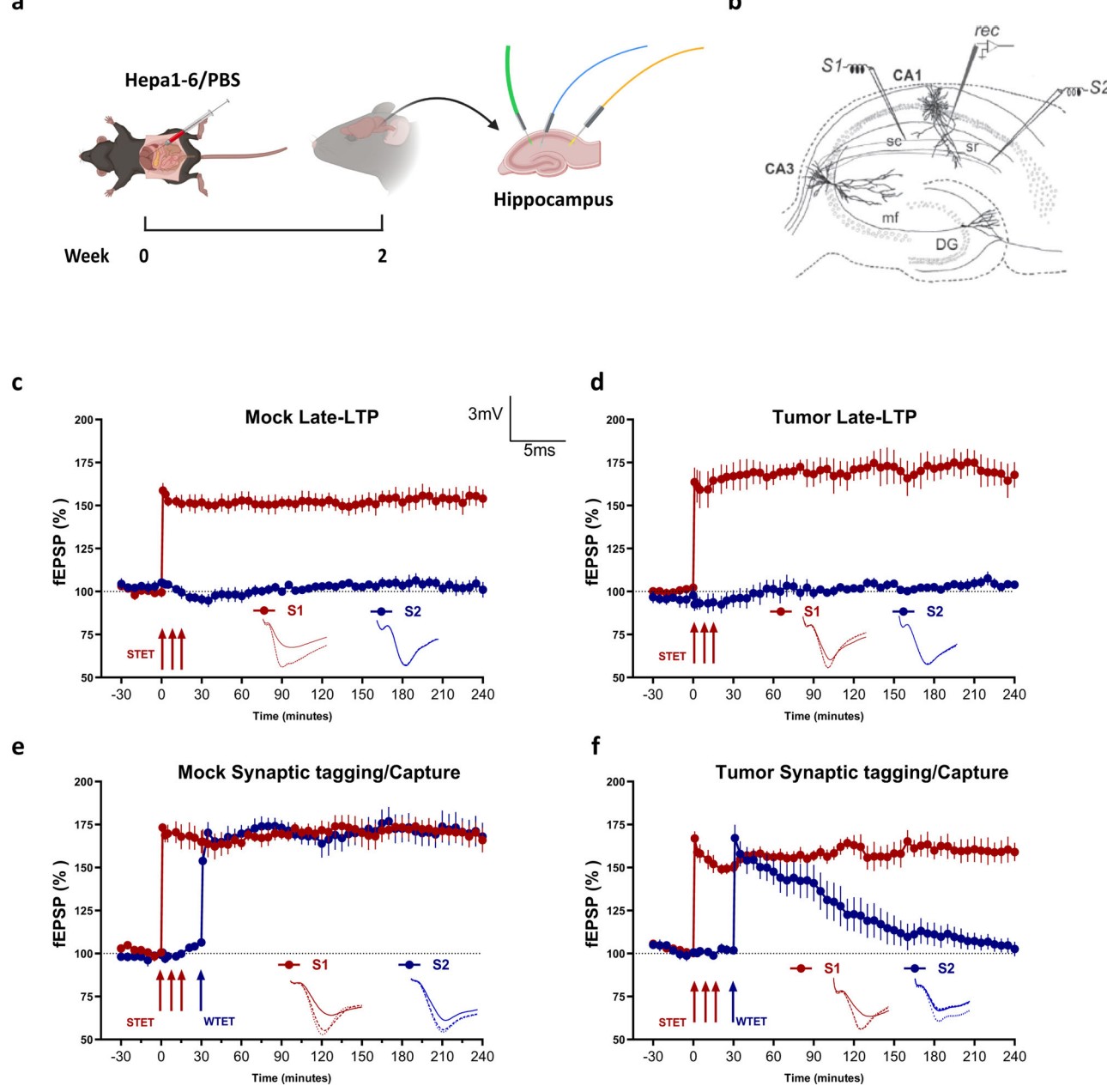

**Fig. 1 Tumor-bearing mice exhibited impaired synaptic tagging and capture. a** Schematic illustration of murine HCC tumor model establishment. Mice were injected intrahepatic with either PBS or Hepa1-6 cells. Two weeks later, mice were killed, hippocampus was removed (Created with Biorender). **b** Schematic representation showing the location of electrodes in the CA1 region of the hippocampus. Recording electrode (rec) was positioned onto CA1 apical dendrites flanked by two stimulating electrodes S1 and S2 in the stratum radiatum (sr) to stimulate two independent Schaffer collateral (sc) synaptic inputs to the same neuronal populations. **c** After recording a stable baseline of 30 min, application of strong tetanization (STET) in S1 synaptic input (red circles) resulted in late-Long Term Potentiation (Late-LTP). The control potentials in S2 (blue circles) were relatively stable throughout the recording in Mock mice ($n = 6$). **d** After recording a stable baseline of 30 min, application of STET in S1 synaptic input (red circles) resulted in Late-LTP. The control potentials in S2 (blue circles) were relatively stable throughout the recording in tumor-bearing mice ($n = 5$). **e** STC was tested using strong before weak (SBW) paradigm. Induction of early-LTP in S2 (blue circles) by WTET 30 min after the Late-LTP induction in S1 (red circles) by STET resulted in Late-LTP in both the synaptic inputs, thereby expressing STC ($n = 6$). **f** The same SBW paradigm when used to test STC in tumor-induced mice failed to express STC ($n = 6$). All data represent mean ± SEM. Representative fEPSP traces shown in each case recorded at −30 min (solid line); 60 min (dotted line); and 180 min (hatched line). Single arrow represents the time point of application of WTET for the induction of early-LTP. Triplet of arrows represent the point of application of STET for the induction of late-LTP. Scale bars for all the traces, vertical: 3 mV; horizontal: 5 ms. sc Schaffer collaterals, sr stratum radiatum, mf mossy fibers, DG dentate gyrus. The experiments were repeated 3 times.

weighted UniFrac analysis (Fig. 2c). PCA analysis also demonstrated that the microbiota community of control and tumor-bearing mice showed clear separation (Fig. 2d). Using linear discriminant analysis (LDA) analysis, we identified differential

bacteria taxa between control and tumor-bearing mice. Specifically, we found that *c_Bacilli and o_Lactobacillales* were enriched in the fecal samples from control mice, while *o_Oscillospiraceae and f_Bacteroidaceae* were enriched in the fecal samples from

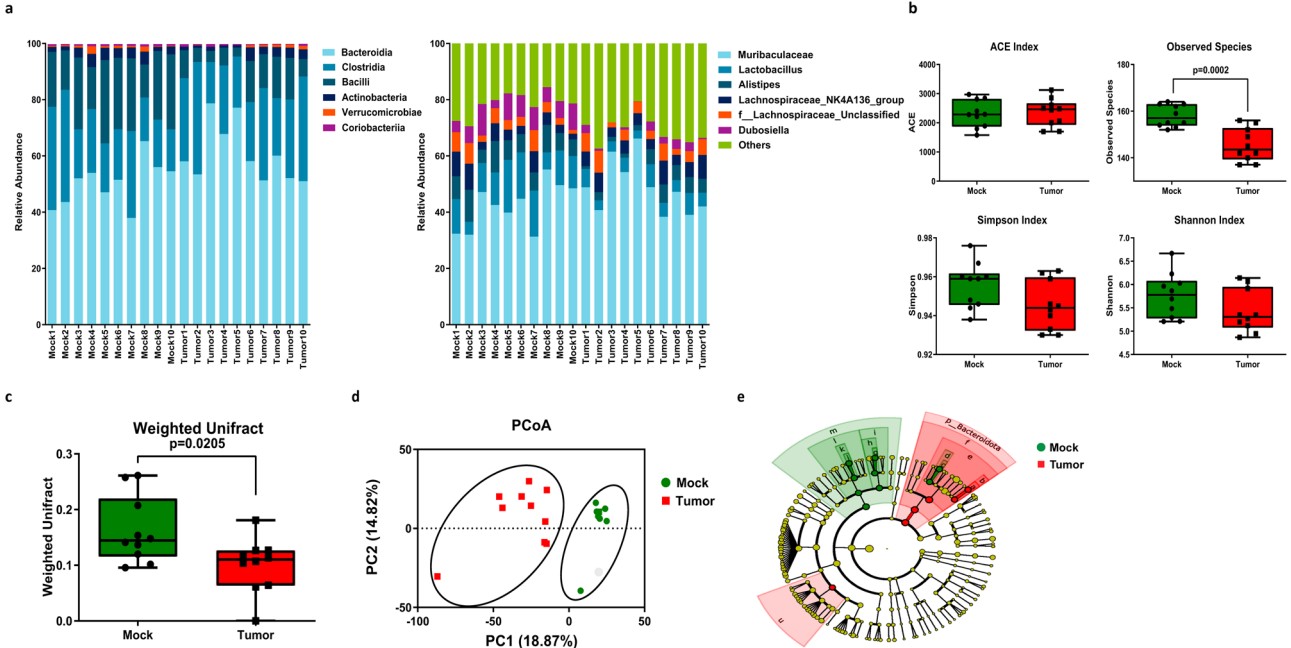

**Fig. 2 HCC tumor-bearing mice exhibited distinct microbiota diversity and composition.** Feces samples were collected from mock and tumor-bearing mice, 16S rRNA sequencing was performed to identify the microbiota species distributions in the samples (n = 10). **a** Average abundance of prevalent microbiota at the class level (left) and genus level (right) in the mock and tumor-bearing mice. **b** Comparison of alpha-diversity using ACE index, Observed species, Shannon index and Simpson index in mock mice compared to the tumor-bearing mice. **c** The Weighted Unifrac was plotted to show the beta diversity. **d** PCoA analysis based on the Unweighted Unifrac was plotted to show the beta diversity. **e** Taxonomic cladogram obtained from LEfSe analysis showing bacterial taxa that were differentially abundant in mock or tumor-bearing mice. Red indicates increased abundance in tumor-bearing mice; green indicates increased abundance in mock mice.

tumor-bearing mice (Fig. 2e). Collectively, our results demonstrated that the microbiota diversity and composition were altered during HCC cancer progression.

**Microbiota depletion restored the cellular associative memory in tumor-bearing mice.** Since the tumor-bearing mice exhibited impaired STC and altered microbiota diversity and composition, we hypothesized that microbiota alteration might account for the plasticity impairments in the tumor-bearing mice. To test this hypothesis, mice were fed with water or ABX for 4 weeks to deplete gut microbiota, followed by orthotopical tumor injection. Two weeks later, mice were killed, liver was removed for tumor measurement, intestine was cut for length measurement, hippocampus was isolated and subjected for LTP and STC analysis (Fig. 3a). The results showed that ABX treatment significantly reduced the HCC progression, as evidenced by decreased tumor size and total liver weight. H&E staining of the tumor section revealed that the liver cells exhibited noticeable degeneration and were loosely arranged. Focal infiltration and scattered presence of inflammatory cells within the lobules, as well as the blood vessel congestion was observed, indicating an inflammatory micro-environment (Fig. 3b). Furthermore, we also observed an elongation of small intestine and cecum after ABX treatment (Fig. 3c). The cecum was also enlarged as previously reported[25]. The hippocampal plasticity analysis showed that the LTP was stable after STET application, suggesting the ABX treatment did not affect LTP (Fig. 3d, e). Additionally, STC was still present in control mice after ABX treatment, suggesting the microbiota had no influence on STC in healthy mice (Fig. 3f). Interestingly, STC was also observed in tumor-bearing mice receiving ABX treatment, suggesting ABX treatment ameliorated tumor-induced impairment in STC (Fig. 3g). Taken together, our results provided evidence that gut microbiota may play a role in tumor-induced

cognitive dysfunction and that antibiotic treatment can ameliorate the associated impairments in STC.

**Tumor-induced microbiota alteration is responsible for STC impairment in hippocampus.** To further confirm whether the impairment in STC is mediated by gut microbiota, we performed microbiota transplantation experiments. Microbiota from control and tumor-bearing mice were transplanted into ABX-treated microbiota-free recipient mice (Fig. 4a). We found that STET induced Late-LTP displayed significantly elevated potentiation and maintained at high level until 240 min in both recipient mice transplanted with either control or tumor-bearing microbiota (Fig. 4b, c). Importantly, WTET also induced a significantly elevated potentiation in recipient mice transplanted with control microbiota and expressed Late-LTP until 240 min, showing intact STC (Fig. 4b). In contrast, recipient mice transplanted with microbiota from tumor-bearing mice failed to express STC. The potentiation in S2 after WTET remained statistically significant only for 60 min, after that the potentiation dropped rapidly, reaching the base level at around 180 min (Fig. 4c). Statistical analysis also revealed that the mean fEPSP after Late-LTP for STC had no significant difference between two different recipient mice (Fig. 4d). Interestingly, the mean fEPSP after early LTP for STC was significantly smaller in recipient mice transplanted with tumor-bearing microbiota compared to recipient mice transplanted with control microbiota after 60 min (Fig. 4e). Taken together, these results demonstrated that tumor-induced microbiota alteration disrupted the expression of STC in hippocampal neurons.

**Tumor-induced STC impairment through microbiota alteration could be partially mediated by IL-1β.** The interactions between microbiota and mammalian immune systems are crucial

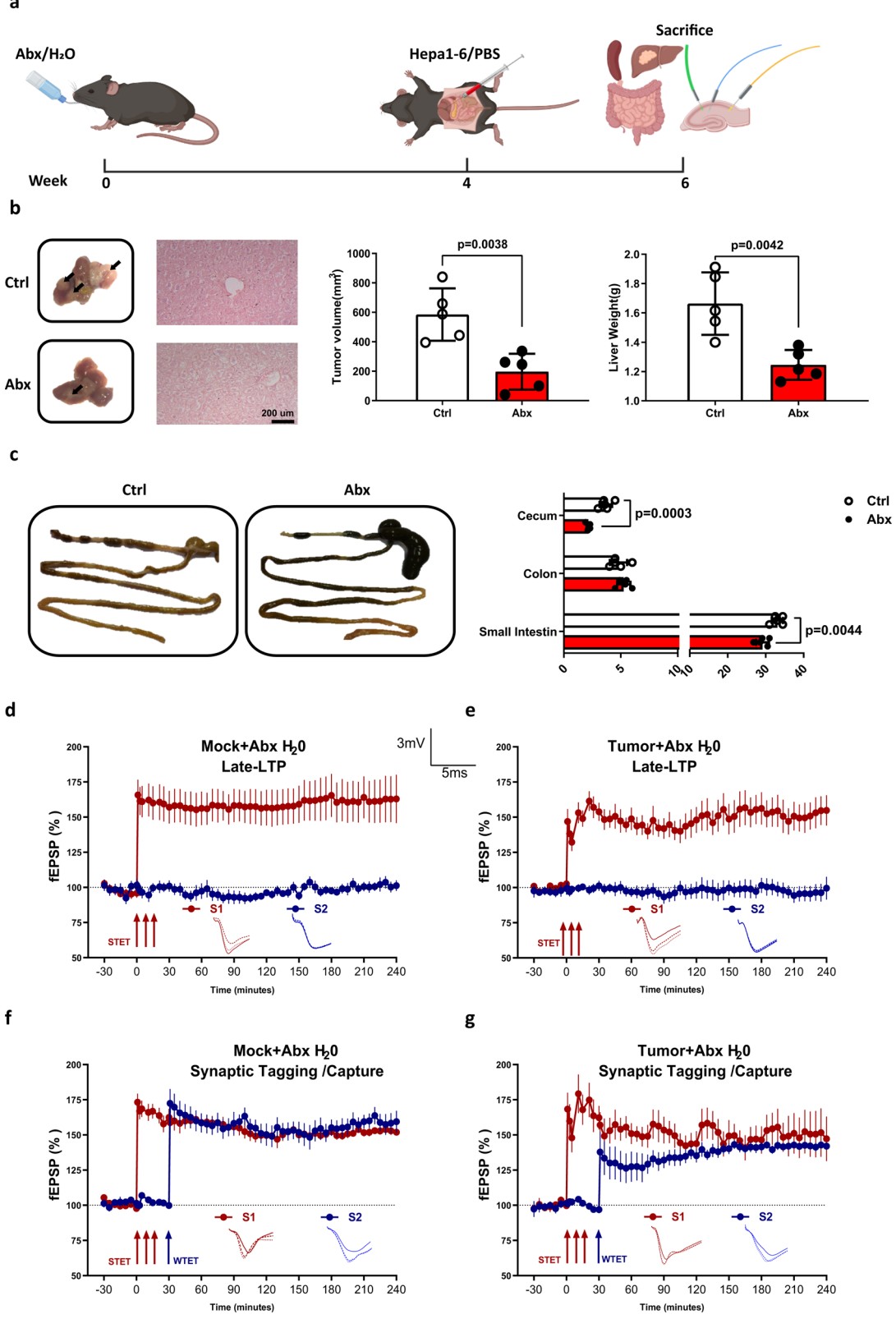

**Fig. 3 Antibiotic treatment ameliorates tumor-induced impairment in STC. a** Schematic illustration of microbiota depletion in HCC tumor-bearing mice. Mice were fed with normal water or ABX water for 4 weeks before establishing the HCC model. Two weeks later, mice were killed, hippocampus was removed (Created with Biorender). **b** The representative of tumor morphology, tumor sizes and liver weight are shown ($n = 5$). **c** The representative of intestine morphology, cecum, colon, and small intestine length are shown ($n = 5$). **d** The Late-LTP was monitored in the mock mice fed with ABX water ($n = 5$). **e** The Late-LTP was monitored in the tumor-bearing mice fed with ABX water ($n = 5$). **f** STC was tested in mock mice fed with ABX water ($n = 5$). **g** STC was tested in tumor-bearing mice fed with ABX water ($n = 5$). Error bars indicate ±SEM. *$p < 0.05$, **$p < 0.01$. The experiments were repeated three times. Symbols, traces, and the scale bar are similar to those shown in Fig. 1.

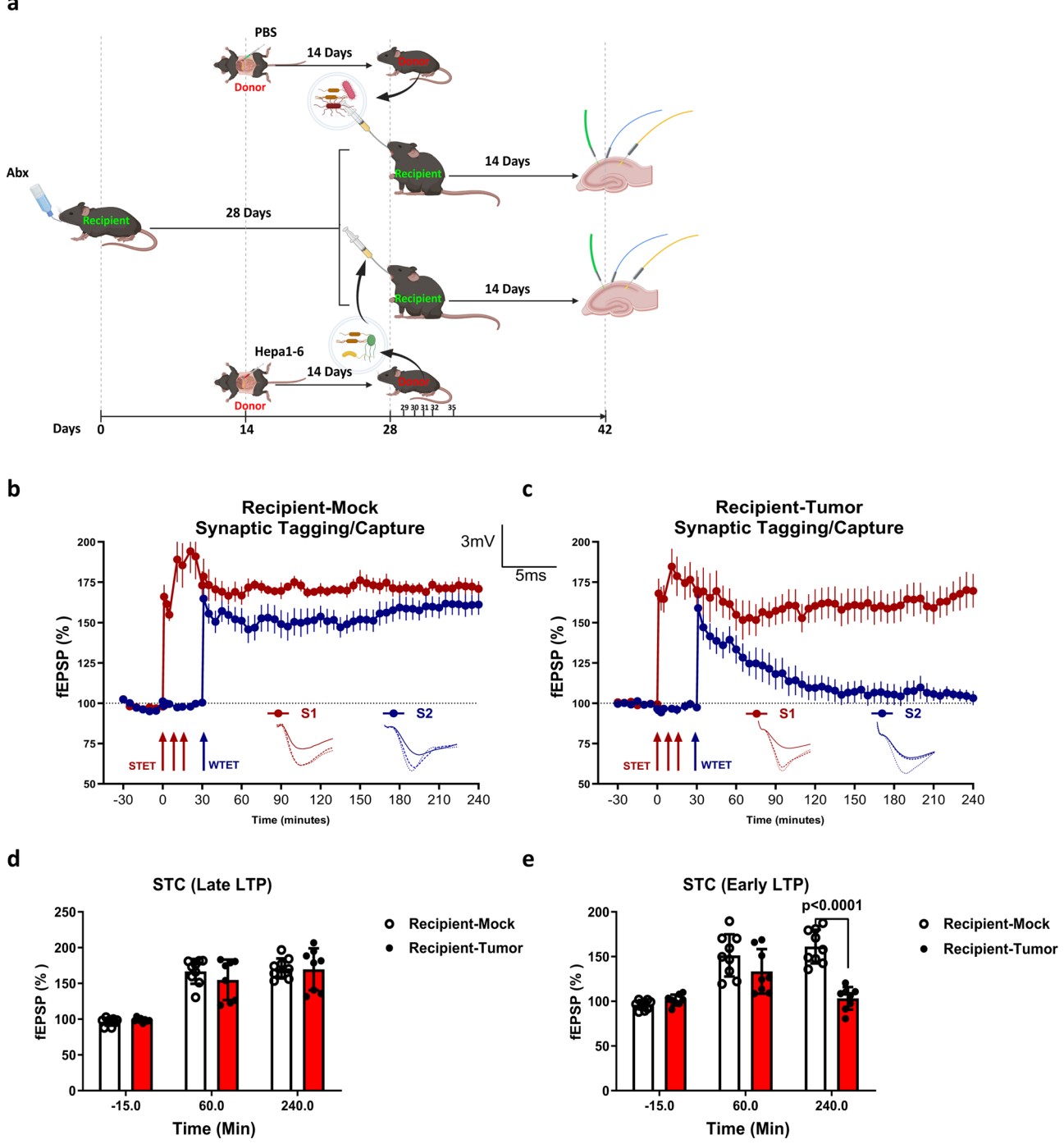

**Fig. 4 Microbiota transplantation from tumor-bearing mice induced deficit in STC. a** Schematic illustration of microbiota transplantation. Recipient mice were fed with ABX water for 4 weeks to deplete the microbiota, followed by oral gavage with fecal contents from either mock or tumor-bearing mice. Another 14 days later, mice were killed, hippocampus was removed (Created with Biorender). **b** STC was tested in recipient mice gavaged with fecal contents from mock mice ($n = 5$–10). **c** STC was tested in recipient mice gavaged with fecal contents from tumor-bearing mice ($n = 5$–10). **d** A histogram of mean fEPSP slope values recorded for recipient mice at three different time points: −15 min (baseline), +60 min, and +240 min after Late-LTP for STC. **e** A histogram of mean fEPSP slope values recorded for recipient mice at three different time points: −15 min (baseline), +60 min, and +240 min after early-LTP for STC. Error bars indicate ±SEM. ***$p < 0.001$. The experiments were repeated three times.

in maintaining the host homeostasis. The microbiota train the host innate and adaptive immunity, while the immune system controls the symbiosis of gut microbiota[26]. Thus, we hypothesized that the altered gut microbiota in the tumor-bearing mice could impair STC in the hippocampus via modulating immune components, especially inflammatory cytokines. To test this hypothesis, multiple inflammatory cytokines in the tumor-

bearing mice fed with control water or ABX water were quantified. Significant reduction of the pro-inflammatory cytokine IL-1β was observed in the serum of the mice in the ABX group. Although other pro-inflammatory cytokines such as IL-6 and IL-17 showed a decreasing trend, the differences were not statistically significant. No differences were observed in the levels of the remaining cytokines, including IL-1α, IL-10, IL-12, IL-23, IL-27,

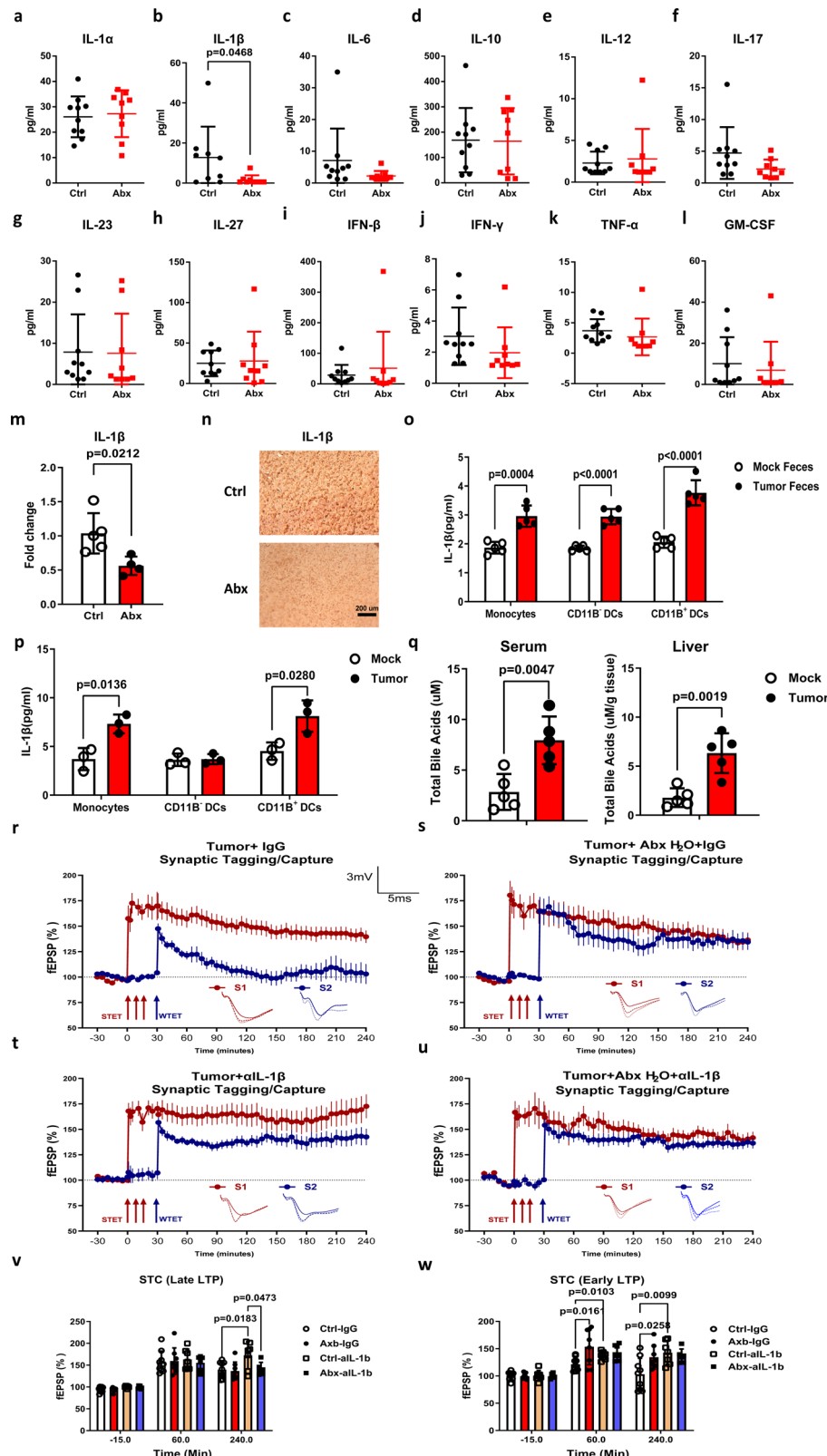

IFN-β, IFN-γ, TNF-α, and GM-CSF (Fig. 5a–l). Furthermore, our study detected elevated serum IL-1β levels in HCC tumor-bearing mice compared to those in wild-type mice (Supplementary Fig. 1), which was in agreement with a prior study showing that HCC patients had increased levels of IL-1β compared to healthy individuals[27]. These observations highlight the potential role of

IL-1β in the development and progression of HCC and support further investigation into the underlying mechanisms driving this association.

Additionally, the *IL-1b* gene expression level was also decreased in the hippocampus of ABX water fed mice compared with control mice (Fig. 5m). Immunohistochemical analysis also

**Fig. 5 Tumor-induced STC impairment was attenuated by IL-1β blocking.** Mice were fed with normal water or ABX water for 4 weeks before establishing the HCC model. Another two weeks later, mice were killed, hippocampus was removed. **a–l** The presence of different inflammatory cytokines, including IL-1α, IL-1β, IL-6, IL-10, IL-12, IL-17, IL-23, IL-27, IFN-β, IFN-γ, TNF-α, and GM-CSF were assessed from the serum of tumor-bearing mice fed with normal or ABX water ($n = 10$). **m** The expression level of IL-1β in the hippocampus of tumor-bearing mice fed with normal or ABX water was quantified by qPCR ($n = 5$). **n** Hippocampus from control or HCC tumor-bearing mice were collected and stained for IL-1β. **o** Intestine LP CD11b⁺CD11c⁻ monocytes, CD11b⁻CD11c⁺ dendritic cells and CD11b⁺CD11c⁺ dendritic cells were sorted and treated with feces contents isolated from control or tumor-bearing mice, the IL-1β level in the supernatant was measured by ELISA ($n = 5$). **p** Intestine LP CD11b⁺CD11c⁻ monocytes, CD11b⁻CD11c⁺ dendritic cells and CD11b⁺CD11c⁺ dendritic cells sorted from Mock or Tumor-bearing mice and stimulated for 3 h, the IL-1β level in the supernatant was measured by ELISA ($n = 5$). **q** The bile acids in the serum and liver tissue were quantified by bile acids kit from Mock and Tumor-bearing mice, the total bile acids level in the liver were standardized with the liver weight ($n = 5$). **r** Mice were fed with normal water or ABX water for 4 weeks before establishing the HCC model. IL-1β neutralization antibody or IgG isotype control was injected into the tumor-bearing mice for 2 weeks. STC was tested in mice fed with normal water and injected with IgG isotype control ($n = 6$). **s** STC was tested in mice fed with ABX water and injected with IgG isotype control ($n = 6$). **t** STC was tested in mice fed with normal water and injected with IL-1β neutralization antibody ($n = 6$). **u** STC was tested in mice fed with ABX water and injected with IL-1β neutralization antibody ($n = 6$). **v** A histogram of mean fEPSP slope values recorded for recipient mice at three different time points: -15 min (baseline), +60 min, and +240 min after Late-LTP for STC. **w** A histogram of mean fEPSP slope values recorded for recipient mice at three different time points: −15 min (baseline), +60 min, and +240 min after early-LTP for STC. Error bars indicate ±SEM. *$p < 0.05$, **$p < 0.01$. The experiments were repeated for three times. Symbols, traces, and the scale bar are similar to those shown in Figs. 1 and 3.

revealed a significant downregulation of IL-1β in the hippocampal tissue of mice in the ABX group compared to the control group (Fig. 5n). The intestinal lamina propria (LP) contains three distinct mononuclear phagocytes based on the different expression patterns of CD11B and CD11C[28]. To further demonstrate the altered microbiota contributed to the enhanced IL-1β secretion, we sorted the LP CD11B⁺CD11C⁻ monocytes, CD11B⁻CD11C⁺ dendritic cells, and CD11B⁺CD11C⁺ dendritic cells from WT mice and stimulated with fecal contents isolated from mock or tumor-bearing mice, respectively (Supplementary Fig. 2). We found that all three cell subsets secreted higher level of IL-1β when stimulated with fecal contents extracted from tumor-bearing mice than from control mice (Fig. 5o), suggesting the tumor-induced microbiota alteration is involved in the upregulation of IL-1β production. In addition, we also found that tumor-bearing mice-derived CD11B⁺ DCs and monocytes secreted higher level of IL-1β compared with those from control mice after being stimulated with LPS, whereas CD11B⁻ DCs showed comparable levels of IL-1β from both control and tumor-bearing mice (Fig. 5p). These results suggested that tumor-induced CD11B⁺ DCs and monocytes are in a hypersensitive stage, which was likely due to the altered microbiome.

Studies have demonstrated the ability of bile acids to promote immune responses during microbiota dysbiosis in disease development[29]. In the present study, we aimed to investigate whether bile acids are associated with increased levels of IL-1β in the HCC model and its potential involvement in the microbiota alteration. The results revealed that the levels of total bile acids were significantly elevated in the serum and liver of HCC tumor-bearing mice compared to control mice (Fig. 5q), suggesting that bile acids could potentially act as pro-inflammatory factors that contribute to the increased levels of IL-1β during HCC progression.

To further confirm the involvement of IL-1β in the impaired STC in the tumor-bearing mice, we performed the same strong before weak STC paradigm in mice receiving IL-1β neutralization antibody. As expected, tumor-bearing mice receiving IgG isotype antibody showed impaired STC (Fig. 5r), while ABX treatment ameliorated tumor-induced deficit in STC (Fig. 5s), evidenced by the increased early-LTP at 60 min and persist until 240 min (Fig. 5v). Importantly, IL-1β neutralization restored the STC in the tumor-bearing mice fed with either normal water or ABX water (Fig. 5t–w). Additionally, αIL-1β treatment further enhanced the Late LTP in tumor-bearing mice compared with mice receiving both ABX water and αIL-1β (Fig. 5v), suggesting ABX treatment also had non-negligible effects on late LTP in the

tumor-bearing mice. Taken together, our results demonstrated that tumor-induced microbiota alteration impaired STC in the CA1 neuron by promoting IL-1β production in vivo.

## Discussion

Over the past three decades, researchers and clinicians have noticed that non-CNS cancer patients may experience cognitive function disorders, including reduced complex processing speed, attention/working memory, and verbal learning efficiency, termed as CRCIs[30]. Traditionally, CRCI symptoms were considered as the neurotoxic effects of cancer therapies such as chemotherapy, radiotherapy as well as immunotherapy[31]. However, emerging evidence suggested that cancer patients may exhibit cognitive decline even before any treatment. Ahles et al. reported that prior to adjuvant treatment, stage 1–3 breast cancer patients were classified as having lower than expected overall cognitive performance as compared to Stage zero patients and healthy controls[32]. Similarly, Simo et al. reported that before chemotherapy, non-small cell lung cancer (NSCLC) patients exhibited cognitive impairments compared with the healthy controls[33]. In addition, Vardy et al. found that localized colorectal cancer patients showed impaired cognitive functions than healthy individuals, with no substantial effect of chemotherapy[34]. Despite this growing evidence, the underlying mechanism by which non-CNS cancer patients, without any cancer therapy interventions, developed CRCIs remains unclear. Recently, Bartels. et al. found that cancer-induced immune autoantibodies against neuronal proteins increased the odds of cognitive impairment in lung cancer patients, suggesting that cancer-induced systemic inflammation might contribute to the brain cognitive function decline[35]. In the liver, hepatic failure often accompanied with brain dysfunction, defined as hepatic encephalopathy (HE), in which increased blood ammonia played a central role in its pathophysiological development[36]. In addition to ammonia, other factors such as short-chain fatty acids, manganese and bile acids have also been reported to contribute to the hepatic encephalopathy development[37]. Among these factors, bile acids have gained considerable scientific interests in recent years. Bile acids are a group of compounds synthesized in the liver and secreted into the digestive tract to aid in the digestion and absorption of dietary fats, vitamins and xenobiotics. Studies have shown that bile acids can disrupt blood brain barrier (BBB) integrity and subsequently expose neurons to potentially toxic molecules. Additionally, bile acids acted as peripheral signaling molecules that activate bile acid receptors present in both BBB and brain, further impacting BBB permeability and brain function[38–40]. In this current study,

we observed an increase in total bile acid levels in the liver and peripheral blood of HCC tumor-bearing mice compared to control mice, indicating that HCC tumor growth affected bile acid secretion and distribution. This finding is consistent with a previous report that increased bile acid promoted IL-1β secretion in DEN-induced murine HCC model[41]. On the other hand, some studies have shown that certain types of bile acids exhibited anti-inflammatory properties by activating farnesoid X receptor and G-protein-coupled bile acid receptor signaling pathways[42]. Further investigation is needed to identify the specific bile acid components present in HCC tumor-bearing mice that may contribute to hippocampal cognitive impairment.

IL-1β played a critical role in BBB dysfunction. IL-1β could activate the cell adhesion molecules such as intercellular adhesion molecule 1 (ICAM-1) and vascular cell adhesion molecule 1 (VCAM-1) in the astrocytes, leading to an increased leukocyte adhesion to the BBB endothelium and their subsequent infiltration into the CNS[43]. It could also promote the secretion of other pro-inflammatory cytokines such as IL-6 and TNF-α to disrupt the paracellular BBB pathway, increasing the BBB permeability that could allow the harmful toxins to enter the CNS[44]. In this current study, we found that HCC tumor-bearing mice had higher serum levels of IL-1β, which could disrupt BBB and allow the bile acids to infiltrate the brain. Whether IL-1β and bile acids could synergistically disrupt BBB permeability warrants further investigation.

Hippocampus plays a critical role in declarative memory formation. Synaptic plasticity refers to the activity-dependent modification of the strength or efficacy of synaptic transmission at synapses, which plays a central role in the formation of learning and memory[45]. Here, we investigated the influence of tumor growth on the hippocampal-dependent plasticity functions. We applied two well recognized models to study the hippocampus memory functions: the long-term potentiation (LTP), and synaptic tagging/capture (STC)[46]. LTP is one of the most comprehensive models of activity-dependent bidirectional synaptic plasticity and has been used to study the long-lasting increase in the efficacy and strength of synaptic transmission[47,48]. STC is a critical model used to describe how late-associativity, cellular mechanisms of associate memory occurs in a time-dependent and input-specific manner[49]. Our results showed that tumor growth had no influence on LTP, as both control and tumor-bearing mice showed persistent LTP after stimulation. Interestingly, we found that hippocampal CA1 pyramidal neurons from tumor-bearing mice failed to express STC. These results suggested that tumor growth disrupted the expression of associative plasticity in hippocampus neurons.

The communication between gut microbiota and neurologic function has emerged as an intriguing pathway that influences the brain memory functions, through multiple mechanisms, including immunological, metabolic as well as endocrine pathways[50,51]. The hippocampus, a critical temporal lobe structure involved in declarative memory formation, plays a vital role in neurodegenerative diseases, such as Alzheimer's disease (AD), Huntington's disease, Parkinson's disease, etc[52]. Clinical studies have demonstrated that gut microbiota diversity and composition are altered in Alzheimer's disease patients compared with healthy individuals, highlighting the importance of gut microbiota on hippocampus functions[53]. In this study, we established a murine orthotopic HCC model by directly injecting Hepa1-6 cells into the liver to investigate the influence of HCC growth on gut microbiota composition and diversity. This orthotopic HCC model is superior to the widely used subcutaneous model as it more accurately recapitulates the complex microenvironmental features of the tumor, including the vasculature, stroma, and surrounding lymphatic system, etc[54]. We found that HCC growth altered the gut microbiota composition and diversity with a significant decrease in Lactobacillus observed in the HCC tumor-bearing mice compared with the control mice. Lactobacillus species were the most widely used probiotics worldwide. In a subcutaneous murine HCC model, administration of Lactobacillus rhamnosus GG has shown to reduce the tumor development[55]. In a DEN-induced rat HCC model, which closely mirrored the human liver cancer development, researchers also found that the relative abundance of Lactobacillus was decreased, indicating a negative association with tumor growth[56]. On the other hand, we observed a significant increase in the relative abundance of Bacteroides in the tumor-bearing mice. Ponziani et al. reported that the relative abundance of Bacteroides were increased in HCC patients, implicating Bacteroides might promote the HCC development[57]. HCC often occurs in the context of chronic inflammation and activates both innate and adaptive immune responses. Studies have suggested that local and systemic immune response induced by hepatocarcinogenesis could disrupt the gut microbiome homeostasis. Moreover, we observed an increase in bile acids in the tumor-bearing mice, which played critical roles in modulating microbiota composition. Bile acids could directly damage the microbiome membrane[58], and alter the microbiota homeostasis through farnesoid X receptor-induced antimicrobial peptides indirectly[59]. Further studies are required to investigate the detailed mechanism of how HCC growth affected the microbiota composition in tumor-bearing mice. Overall, these findings proved that HCC growth altered the microbiota composition in a manner that favored tumor progression, with a decrease of good bacteria and an increase in bad bacteria. However, it is important to note that the extent and specificity of these changes may vary depending on the particular tumor model being investigated. Additional studies are needed to determine whether these findings can be generalized across different HCC models. Nevertheless, the similarities overserved in the present study and in other HCC models suggested that these results may have broader implications and warrant further investigations.

Alpha and beta diversity are two metrics to measure the complexity of microbiota. Alpha diversity describes the richness in a given sample, while beta diversity is used to define the extent of absolute or relative overlap in shared taxa between samples[60]. Studies have suggested that gut microbiota dysbiosis was associated with the therapeutic outcomes in various cancer types[61]. Gut microbiota diversity has been identified as an independent predictor of survival in cervical cancer patients receiving chemoradiation[62]. In non-small cell lung cancer, renal cell carcinoma, or urothelial carcinoma patients, higher gut microbiota diversity was associated with increased lymphocytes infiltration into the tumor[63]. In human, the Shannon index and Simpson index of the microbiota extracted from fecal samples were decreased in the nonalcoholic steatohepatitis-associated HCC patients compared with healthy control[64]. Additionally, beta diversity analysis has also identified distinct clustering of microbiota species between HCC patients and healthy controls[65]. In our murine HCC model, we observed a decreased alpha diversity in the tumor-bearing mice, reflected by lower number of observed species. Beta diversity analysis further demonstrated distinct clustering of the microbiota between tumor-bearing and control mice. These similar findings between our murine HCC samples and clinical data suggested that gut microbiota underwent a shift from high diversity to low diversity during cancer progression, indicating that microbiota diversity might play an essential role in tumor development. However, more evidence is needed to further elaborate the exact function of gut microbiota diversity in carcinogenesis.

Microbiota alteration induced by lifestyle, diet, diseases, antibiotic treatment, or other factors plays an important role in

human brain disorders. Metabolites and other molecules released by microorganism trigger host immune response, resulting in cytokine secretion and inflammation in the CNS, which further contributes to the pathogenesis of host brain disorders, including stress, depression, Alzheimer's diseases, etc. Hence, we investigated whether the altered microbiota contributed to the impaired STC in the tumor-bearing mice and found that sterilization of gut microbiota restored the STC in the tumor-bearing mice. This result further demonstrated that tumor-induced microbiota alteration was responsible for the associative plasticity impairments observed in the hippocampus.

Clinically, there are no guidelines for treating CRCIs[66]. However, with increasing awareness of the gut microbiota's impact on brain function, several intervention approaches, including fecal microbiota transplantation, antibiotics therapy, or probiotics treatments are undergoing clinical trials for hippocampus-related brain disorders[24]. Our study showed that transplanting the microbiota from healthy mice could restore the STC, whereas transplanting the microbiota from tumor-bearing mice still failed to induce a persistent STC. These findings provided evidence that microbiota transplantation could serve as a promising therapeutic approach for hippocampus-related brain cognition impairment.

The interaction between gut microbiota and brain has been extensively studied in the past decade. However, the detailed mechanism of how microbiota modulates the brain function is poorly understood. One hypothesis suggests that gut-associated immune system may influence the neuron functions. For example, Wang et al. reported that during Alzheimer's disease progression, the altered gut microbiota composition induced pro-inflammatory Th1 cells differentiation and proliferation, which subsequently activated the M1 microglia, contributing to Alzheimer's disease-associated neuroinflammation[67]. In the hippocampus, chronic overexpression of IL-1β could decrease neurogenesis, resulting in selective cognitive dysfunction[68]. Conditional deletion of IL-1R1 in hippocampal glutamatergic neurons abrogated the stress-induced deficits in social interaction and working memory in mice[69]. In our HCC model, we found that sterilization of the gut microbiota significantly reduced serum IL-1β level. We also observed higher IL-1β level in the hippocampus in HCC tumor-bearing mice, suggesting tumor-induced microbiota alteration might affect the hippocampal STC through IL-1β signaling. Blocking IL-1β signaling using neutralization antibody restored the STC in the tumor-bearing mice, demonstrating the critical role of IL-1β in hippocampus cognitive dysfunction.

IL-1β is a critical cytokine that modulates both innate and acquired immune responses. Microbiota dysbiosis has been reported to be associated with increased IL-1β secretion. Microbiota-driven IL-1β release by intestinal macrophages could regulate the Csf2 production by RORγt+ ILC3 to maintain the intestine homeostasis[70]. Furthermore, microbiota-induced IL-1β is also essential for the development of steady-state TH17 cells and regulatory B cells in the intestine[28,71]. We observed elevated IL-1β production in tumor-bearing mice with microbiota alteration, although the detailed mechanism remains unknown. It is possible that tumor-induced Lactobacillus reduction may partially contribute to the elevated IL-1β secretion, as several Lactobacillus species have been implicated to modulate pro-inflammatory responses. For instance, Lactobacillus rhamnosus GG could prevent colitis relapse in antibiotic treated rat by downregulating IL-1β production[72]. Lactobacillus acidophilus could inhibit Porphyromonas gingivalis-driven IL-1β secretion[73]. Thus, decreased Lactobacillus abundance disrupts gut homeostasis, and results in the upregulated IL-1β production. In addition to IL-1β, a decreasing trend of the pro-inflammatory cytokines IL-6 and IL-17 was also observed in the ABX group mice. These cytokines have been implicated in the gut-liver axis during liver pathogenesis. For example, altered gut microbiota-derived short-chain fatty acid have been shown to suppress IL-17 expression by γδ T cells, which are critical in the development of HCC[74,75]. Tans-signaling of IL-6 has also been found to be essential for HCC progression[76]. Notably, single depletion of IL-1β with the neutralization antibody could sufficiently restore the STC in the tumor-bearing mice, further ABX treatment failed to rescue the STC to a higher extend, suggesting that IL-1β is the dominant pro-inflammatory cytokine mediating STC decline in the tumor-bearing mice.

In conclusion, our study sheds new light on tumor-induced plasticity dysfunction in the hippocampus via the liver/tumor-gut-immune-brain axis. We demonstrated that HCC growth disrupted the gut microbiota homeostasis, resulting in elevated IL-1β production, which subsequently impaired the STC in the hippocampus. Antibiotics or probiotic treatments, as well as IL-1β blocking could serve as promising therapeutic strategies for treating cancer-related cognitive impairment.

## Methods

**Mice**. Male C57BL/6 mice (6–8 weeks old) were purchased from Invivos Pte Ltd. (Singapore). All animal studies were approved by the National University of Singapore Institutional Animal Care and Use Committee under protocol number R15-1041. Mice were housed in a temperature-controlled (22–24 °C) and humidity-controlled specific pathogen-free facility on a 12 h light/dark cycle with access to water and standard diet.

**Cell culture**. Murine HCC cell line Hepa1-6 was purchased from American Type Culture Collection (ATCC, Manassas, VA). Cells were cultured in DMEM medium (Hyclone, Piscataway, NJ), containing 10% FBS (Hyclone), 10 mmol/L HEPES (Hyclone), 1 mmol/L sodium pyruvate (Hyclone), MEM nonessential amino acids (Life Technologies, Carlsbad, CA), and 1% penicillin–streptomycin antibiotics (Sigma–Aldrich, St. Louis, MO). Cells were cultured in T75 flasks (Corning, NY) and incubated at 37 °C with 5% $CO_2$.

**Murine HCC models**. Six-weeks-old C57BL/6 mice were orally administrated of an antibiotic cocktail, consisting of ampicillin (0.5 g/L), vancomycin hydrochloride (0.25 g/L), neomycin trisulfate salt hydrate (0.5 g/L), and metronidazole (0.5 g/L) in 1% sugar water for four weeks. HCC model was established in the respective mice via surgical orthotopic implantation. $1 \times 10^6$ Hepa1-6 cells in 25 μl PBS or PBS alone were injected into the mouse liver directly. A control group was included and fed with 1% sugar water throughout the treatment. The mice were euthanized two weeks post-surgery and the livers and intestinal tract were harvested for analysis. Tumor sizes were measured with digital callipers and calculated by the following formula: tumor volume = $0.5 \times$ width$^2 \times$ length.

**Hippocampal slice preparation**. Mice were euthanized with $CO_2$ and then decapitated briefly. Brains were removed quickly and cooled in 4 °C artificial cerebrospinal fluid (ACSF). The ACSF consists of 124 mM NaCl, 4.9 mM KCl, 1.2 mM $KH_2PO_4$, 2.0 mM $MgSO_4$, 2.0 mM $CaCl_2$, 24.6 mM $NaHCO_3$, and 10 mM D-glucose, equilibrated with 95% $O_2$, 5% $CO_2$ (32 L/h). The pH of aCSF was between 7.3 and 7.4 when bubbled with 95% oxygen and 5% carbon dioxide. To obtain transverse hippocampal slices of 400μm in thickness, a manual tissue chopper (Stoelting, Wood Dale, Illinois) was used to slice the hippocampus and the slices were then incubated at 32 °C in an interface chamber (Scientific Systems Design, Ontario, Canada))[77]. To ensure the optimal condition of hippocampal slices for electrophysiology studies, the complete procedure encompassing animal dissection, hippocampal slice preparation, and placement of slices onto the chamber was executed efficiently, taking ~5 min.

**Immunohistochemistry and histopathology**. Following euthanasia, tumor tissues were removed aseptically and immediately fixed in 4% formalin at room temperature. The fixed tissues were processed through graded concentrations of ethanol and xylene and were then embedded in paraffin wax. Tissue sections of 4–5 mm were mounted on adhesive glass slides and were stained with H&E. Hippocampal sections fixed and then deparaffinized and treated with 0.08% $H_2O_2$ for 30 min to block endogenous peroxidase. Slides were incubated with goat IL-1β (R&D Systems, MN, MN) at 4 °C overnight, followed by incubation with HRP-conjugated donkey anti-goat IgG. Diaminobenzidine was used to develop the staining reaction, and nuclear counterstaining was performed with haematoxylin. Slides were coded and examined by a pathologist who was blinded for the experimental history of the animals.

**Electrophysiology**. In all the electrophysiology recordings, two-pathway experiments were performed. Two monopolar lacquer-coated stainless-steel electrodes (5 MΩ; AM Systems, Sequim) were positioned at an adequate distance within the stratum radiatum of the CA1 region for stimulating two independent synaptic inputs S1 and S2 of one neuronal population, thus evoking field excitatory post-synaptic potentials (fEPSP) from Schaffer collateral/commissural-CA1 synapses (Fig. 1b). Pathway specificity was tested using the method described in[78]. A third electrode (5 MΩ; AM Systems) was placed in the CA1 apical dendritic layer for recording fEPSP. After the pre-incubation period of 3 h, a synaptic input-output curve (afferent stimulation vs. fEPSP slope) was generated. Test stimulation intensity was adjusted to elicit fEPSP slope of 40% of the maximal slope response for both synaptic inputs S1 and S2. The signals were amplified by a differential amplifier, digitized using a CED 1401 analog-to-digital converter (Cambridge Electronic Design, Cambridge, UK) and monitored online with custom-made software. Late-LTP was induced using strong tetanization in the S1 synaptic input consisting of three stimulus trains of 100 Hz, 100 pulses (strong tetanus, 100 Hz; duration, 0.2 ms per polarity; intertrain interval, 10 min). Synaptic tagging and capture method involves induction of late-LTP in S1 followed by and Early-LTP in S2 within 30 min interval. Early-LTP was induced using a weak tantalization consisting of one 100-Hz train (21 biphasic constant-current pulses: pulse duration per half-wave, 0.2 ms)[77].

**Reverse transcription and qualitative polymerase chain reaction**. Total RNA extraction of hippocampus samples was performed using the RNeasy Mini Kit (Qiagen, Hilden, Germany). RNA concentrations of the samples were firstly quantified using the NanoDropTM spectrophotometer 24 (Thermo Fisher, Waltham, MA), followed by the reverse transcription and qualitative polymerase chain reaction. 1 μg of RNA was reverse transcribed to cDNA using the Maxima First Strand cDNA Synthesis Kit for q-PCR (Thermo Scientific) on the PCR machine (BioRad, Hercules, CA). qPCR was then performed using GoTaq qPCR Master Mix (Promega, Madison, WI) on the ABI 7500 Real-Time PCR system (Applied Biosystems, Waltham, MA). 10 ng of cDNA, 0.2 μM primers and GoTaq qPCR Master Mix was used for each reaction and three replicates were performed for each gene. To analyze the data, the threshold cycles depicting the gene expression levels were normalized to GAPDH, a housekeeping gene (ΔCt). This is then followed by the normalization to the control group (ΔΔCt), to achieve the final fold change in gene expression. Primers used were: GAPDH forward primer: 5′-CATCACTGCCACC CAGAAGACTG-3′, GAPDH reverse primer: 5′- ATGCCAGTGAGCTTCCCGT TCAG-3′; IL-1b forward primer: 5′-CACAGCAGCACATCAACAAG-3′, IL-1b reverse primer: 5′-GTGCTCATGTCCTCATCCTG-3′.

**16S rRNA sequencing**. Mouse feces were collected into sterile EP tubes. 100 mg of feces samples were homogenized on a FastPrep-24™ Classic bead beating grinder and lysis system (MP Biomedicals, Irvine, CA). Feces contents were pelleted by centrifuging at 10000 rpm for 10 min. The supernatant was collected and total DNA was extracted with the FastDNA spin kit (MP biomedicals, Solon, OH). Primers 338F (ACTCCTACGGGAGGCAGCAG) and 806R (GGACTACHV GGGTWTCTAAT) were used to amplify the V3–V4 domain of the 16S rRNA. Sequencing 16S rRNA was performed on an Illumina MiSeq platform (Azenta, Beijing, China).

**Microbial analysis**. The data were processed, two sequences of each read pair were merged according to overlapping sequences. The read merge is deemed to be successful only if the overlapping sequence is least 20 bp long. After merging, undetermined bases were removed from the resulting sequence. Adapter sequences and primer sequences were removed using Cutadapt (version. 1.9.1)[79]. Each set of paired-end sequence reads was assembled and demultiplexed with QIIME (version. 1.9.1)[80]. Unique sequences are extracted from the optimized sequences with the read count information. All optimized sequences are compared with OTU representative sequences, and sequences of >97% similarity to a specific OTU representative using the VSEARCH program (version 1.9.6)[81] against the Silva 132 database[82]. For taxonomic classification, the Ribosomal Database Program (RDP)[83] classifier was employed with a confidence threshold of 0.8, utilizing the Silva 132 database that offers taxonomic categories predicted at the species level. The obtained results from the OTU analysis, including the ACE index, Observed species, Shannon index, Simpson index, coordinate analysis (PCoA) and Weighted Unifrac were calculated and compared using vegan package[84] in R (version. 3.3.1). LEfSe analysis was performed using the Galaxy implementation of LEfSe[85].

**Fecal transplantation**. Six-weeks-old recipient C57BL/6 mice were orally administered of an antibiotic cocktail, consisting of ampicillin (0.5 g/L), vancomycin hydrochloride (0.25 g/L), neomycin trisulfate salt hydrate (0.5 g/L), and metronidazole (0.5 g/L) in 1% sugar water for four weeks. Two weeks before fecal transplantation, eight-weeks-old donor C57BL/6 mice were given intrahepatic PBS or Hepa1-6 injection to establish the murine HCC model. Two weeks later, fecal pellets were collected from the donor HCC tumor-bearing mice or control mice. Feces pellets were mashed with the FastPrep-24™ Classic bead beating grinder and lysis system (MP Biomedicals), and resuspended in 2 ml saline solution to get a concentration of 60 mg fecal material per 200 μl saline. Recipient mice were orally gavaged with 60 mg fecal material. The gavage was performed on day 1, 2, 3, 4, 7 for a total of five doses. Another one week later, recipient mice were killed.

**In vivo IL-1β neutralization**. To block IL-1β in vivo, 200 μg Armenian Hamster anti-Mouse IL-1β neutralization antibody (Bio X Cell, Lebanon, NH) was i.p. injected into the mice one day prior the tumor inoculation. The antibody was given three times per week. Armenian Hamster IgG was used as isotype control.

**Cytokine analysis**. Mouse serum sample was prepared by centrifuging the blood at 3000 rpm for 30 min at room temperature. The top transparent serum layer was collected into sterile EP tubes. Serum cytokine levels were assessed by LEGEN-Dplex Multiplex assays (Biolegend, San Diego, CA) by BD LSRFortessa™ X-20 cytometry (BD Biosciences, San Jose, CA) and analyzed by LEGENDplex™ Data Analysis Software Suite (Biolegend). The kit includes mouse IL-1α, IL-1β, IL-6, IL-10, IL-12, IL-17, IL-23, IL-27, IFN-β, IFN-γ, TNF-α, and GM-CSF. The standard curve and the samples were measured using an BD FACSAria™ Fusion Flow Cytometers (BD Bioscience, Franklin Lakes, NJ, USA). The results were analyzed with the LEGENDplex™ Data Analysis Software Suite from Qognit (https://legendplex.qognit.com/user/login?next=home), which could differentiate specific beads for each analyte based on their size and fluorescence intensity, where 4000 events had to be acquired per analyte. The concentration of each analyte was quantified based on the individual standard curve of each parameter.

In some experiments, the IL-1β expression was detected by mouse IL-1β ELISA kit (Biolegend).

96-well plate (Corning) was pre-coated with IL-1β capture antibody and incubated at 4 °C overnight. The next day, wells were washed and blocked with PBS containing 2% FBS for 1 h at room temperature. Standards and serum samples were added into the wells and incubated for 2 h at room temperature. IL-1β detection antibody was added into each well and incubated for another 1 h. The streptavidin-HRP was added into each well for 20 min. Finally, substrate solution was added and incubated for 20 min in the dark. The reaction was stopped by adding stop solution, and the optical density of each well was detected and measured on a photospectrometer (Bio-Rad, Hercules, CA) at a wavelength of 450 nm. The concentration of IL-1β in the culture supernatants was calculated according to the standard curve.

**Total bile acids quantification**. 100 mg of mouse liver was homogenized in cold isopropanol and supernatant was collected. The total bile acids in the serum and liver were measured using Total Bile Acid Assay Kit (MyBioSource, San Diego, CA) following the manufacturers' instructions. Samples were incubated with 3α-HSD, NADH, and thio-NAD$^+$. Thio-NAD$^+$ is converted to its reduced form Thio-NADH by bile acids. Each Bile Acid standard and sample is measured in triplicate. A fresh standard curve is prepared for each assay. Diluted bile acid standards or samples (20 μL) are added to a 96-well plate. Assay Reagent A (150 μL Thio-NAD$^+$) is added to each well and mixed thoroughly, followed by a 5-minute incubation at 37 °C. For enzyme treatment, Assay Reagent B is added (50 μL 3α-HSD and NADH) to the standards and half of the paired sample wells, while the other half of the paired sample wells receive NADH Reagent (50 μL). The plate is incubated at 37 °C for 30–60 min. Finally, the plate is read using a microplate spectrophotometer (Bio-Rad, Hercules, CA) at primary (405 nm) and secondary (630 nm) wavelengths. To calculate the results, subtract the absorbance at 630 nm from the absorbance at 405 nm to obtain the net absorbance. Next, determine the average absorbance values for each sample and standard. Graph the standard curve using the corrected standard values. Then, subtract the absorbance values of the sample wells without 3α-HSD from the absorbance values of the sample wells treated with the enzyme (3α-HSD and NADH) to obtain the absorbance difference (ΔA). This difference represents the activity of the enzyme 3α-HSD. Finally, compare the change in absorbance (ΔA) of each sample to the standard curve to determine the quantity of bile acid present in the sample.

**Isolation of lamina propria leukocytes**. Mice were killed and the small intestine was removed by cutting below the stomach and above the caecum. Intestinal contents were cleared by flushing with PBS. Intestines were cut into small pieces and put into 40 ml digestion buffer containing 5 mM EDTA, 1 mM DTT, 0.2 g dispase, and then shake for 40 min at 200 rpm, 37 °C. After incubation, filter the cell solution through a 40 μm cell strainer and pellet the cells by centrifugation. The cells were further applied with percoll gradient separation to isolate the leukocytes. Cells were stained with DAPI (Thermo Scientific), BUV395-anti-mouse CD45 (Clone: 30-F11, BD Bioscience), PerCP/Cy5.5-anti-mouse CD11c (clone: N418, Biolegend), PE-Cy7-anti-mouse CD11b (clone: M1/70, Biolegend) for 30 min before sorted on the BD FACSAria™ Fusion Flow Cytometers (BD Biosciences).

Feces from control or tumor-bearing mice were suspended in sterile PBS (100 mg/ml), homogenized, and filtered through 40 mm cell strainer to remove aggregates and used to stimulate the LP phagocytes at a 1:200 dilution for 4 h. After that, the medium was replaced with fresh medium containing 100 mg/ml gentamicin and continued culture for 16 h. The supernatant was collected for IL-1β measurement.

**Statistics and reproducibility**. All data are represented as mean ± SEM. The fEPSP slope value expressed as percentages of average baseline values per time point was subjected to statistical analysis using GraphPad Prism 6.0 (GraphPad, San Diego, CA, USA). Nonparametric tests were used as the normality variations at small sample sizes. Wilcoxon signed rank test (Wilcox test) was used to compare within one group and Mann–Whitney $U$ test (U test) was used when data were compared between groups. Statistical comparisons for tumor size and immunophenotyping were performed using Student's $t$ test between two groups and one-way ANOVA for multiple groups. $P$ value less than 0.05 was considered as statistically significant. The significance levels are marked *$P < 0.05$, **$P < 0.01$, and ***$P < 0.001$. All sample sizes are provided in the figures and figure legends.

**Reporting summary**. Further information on research design is available in the Nature Portfolio Reporting Summary linked to this article.

## Data availability

The numerical source data behind the graphs in the paper can be found in Supplementary Data. The 16S sequencing data that support the findings has been deposited to Genome Sequence Archive (GSA) under Accession ID: CRA009942.

## Code availability

The code used in the analysis of 16S sequencing data is available at https://github.com/UmeAmee/16s-seq.

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

## Acknowledgements
We thank Dr. P. Hutchinson and Mr. G. Teo (NUS Immunology Program Flow Cytometry Core Facility) for assistance with FACS sorting. We thank Ms. Lee Kang Yi for her assistance with the H&E and IHC staining. This work is supported by the Ministry of Health grant MOH-000641-00 (S.S.), Ministry of Education Academic Research Fund Tier 3 MOE2017-T3-1-002 (S.S.), NUSMED-FOS Joint Research Programme NUHSRO/2018/075/NUSMed-FoS/01 (S.S.), NUHS seed fund NUHSRO/2020/145/RO5+6/Seed-Sep/05 (S.S.), and National Research Foundation of Singapore (NRF-CRP26-2021RS-0001) (H.L.).

## Author contributions
Y.Z., Y.M., H.Y.T., Z.B.H., and S.N.M.S. performed experiments. Y.Z., Y.M., and N.B. analyzed the data. Y.Z. and Y.M. wrote the manuscript. S.S. and H.L. designed experiments and provided insightful advice and critiques and edited the manuscript.

## Competing interests
The authors declare no competing interests.
