## [Peer Review File · Communications Biology]

Reviewers' comments:

Reviewer #1 (Remarks to the Author):

In this article, the authors are showing in a HCC model, that the tumor is mediating impairment of cognitive function by modulating the gut microbiota population leading to an upregulation of IL-1 β . While previous studies have shown that the gut microbiota can have an impact on HCC tumor development, this study provides evidence showing that the tumor can also have an impact on the gut microbiota by altering it. Studies are well performed and the manuscript is well written.

Major comments:

1) The authors are showing in figure 3B the representative tumor morphology, tumor volume and liver weight. Can the authors add H&E stained sections of liver from ABX and control mice to Fig.3B?

2) In figure 5, the authors are showing that the depletion of the microbiota in tumor-bearing mice lowers the amount of IL-1 β in the serum and in the hippocampus. However, there is no evidence showing that the level of IL-1 β in the serum of tumor-bearing mice is higher than in the control group. Have the authors measured the amount of IL-1 β in the serum of both groups (figure 1)?

3) In Figure 5, the authors are showing that intestine LP leukocytes incubated with feces from tumor-bearing mice are producing more IL-1 β than when incubated with a healthy microbiota. The amount of IL-1 β produced by Intestine LP leukocytes in vivo should also be measured. This could be assessed by isolating the intestine LP from mice harboring tumors (vs. the control group) to determine by flow cytometry the amount of IL-1 β produced by each cell population (CD11B+CD11C-, CD11B-CD11C+, CD11B+CD11C+).

4) In figure 5, the authors are showing a significant reduction of IL-1 β in the serum and in the hippocampus of the ABX group mice compared to the control group. Does the IL-1 β amount present in the hippocampus have also been assessed by IHC in both groups?

5) The authors should discuss more about by which mechanism HCC tumor can modulate the gut microbiota population.

Minor Comments:

1) The number of mice used for each experiment should be indicated in each figure.

2) There is no mention of the Figure 3A in the text.

3) For the Figure 5N, the name of each immune cell type should be mentioned in the figure or in the legend, and the gating strategy used to isolate each cell population should be provided as supplementary information.

Reviewer #2 (Remarks to the Author):

In this study, Mei and colleagues used a preclinical model to evaluate the role of the gut microbiota in mediating the cognitive disruption that is commonly observed in patients with neoplastic lesions. They used a mouse model of hepatocellular carcinoma to evaluate the contribution of the gut microbiota to changes in the electrophysiology recordings as surrogate of synaptic plasticity. Using, antibiotic-mediated depletion of the gut microbiota as well as fecal transplants in antibiotic-treated mice,

authors elegantly demonstrated the involvement of the gut microbiota in the observed changes in synaptic tagging/capture recordings of mice with induced hepatocellular carcinoma. Authors also demonstrated that these physiological alterations were likely mediated by the microbiota inducing IL1b production in the gut of animals with hepatocellular carcinoma. Overall, the paper is well-written, the methods used are sound and the research rigorous with no overstatement. I have a few suggestions for the authors:

- IL1b has been shown to impact blood brain barrier, however this is not considered by the authors in the discussion. I believe this is an important point to address as blood brain barrier leakage would facilitate the translocation of toxic byproducts that can further exacerbate the observed phenotype.
- Just wondering whether the authors have considered bile acid metabolism as mediator of both the initial changes in the gut microbiota of mice with hepatocellular carcinoma as well as the observed cognitive phenotype. Altered bile acid metabolism is associated with altered liver function. High levels of bile acids in the gut are considered proinflammatory and have known antimicrobial effects. On the other hand, some antibiotics are known to induce cholestasis, which would retain bile acids in the liver.
- In figure 2A-B, please add the profiles of all the animals rather than the averaged abundance. Also Figure 2D and 2E are redundant as they show the same effect using different methods. Please keep just one.
- Figure 5A-L. Are differences for any inflammatory marker significant? IL6 and IL17 behave in the same way as IL1b, but they are not further considered by the authors. Thoughts?
- Please carefully check the manuscript for typos etc. E.g. Line 217 tumor-nearing should be tumor-bearing(?)
- Sequencing data should be deposited in a public repository (e.g. SRA).

Reviewer #3 (Remarks to the Author):

The paper by Mei and Colleagues is a study on a very hot topic, related to the effects of gut microbiota in the brain, in the presence of a tumor-induced state of inflammation.

The study is well organized and different experimental approaches are used to verify the principal hypothesis of a role for IL-1b as mediator of the gut-brain communication in a model of hepatocellular carcinoma.

However, many weak points need to be addressed before considering the manuscript suitable for the publication.

Major points:

- 1) English must be deeply revised
- 2) More than one cell line should be used in the animal models.
- 3) Too low number of mice is used for the microbiome analyses. Acceptable numbers range from 10 to 20.
- 4) Weak statistical analyses. The authors never mentioned the number of samples/repeats performed.
- 5) The only cytokine modified by Abx treatment is IL1b, as shown in fig 5. Is the difference in IL1b still significant if the outlier in control is eliminated?

We would like to thank all the reviewers for reviewing our manuscript and providing constructive comments, which greatly help us improve the manuscript. Additional experiments have been performed as suggested by the reviewers, and the results are incorporated into the manuscript. They are described in detail in the following point-to-point session. We hope the manuscript can be considered for publication in its revised version.

Comments from the Reviewers:

Reviewer #1:

In this article, the authors are showing in a HCC model, that the tumor is mediating impairment of cognitive function by modulating the gut microbiota population leading to an upregulation of IL-1 β . While previous studies have shown that the gut microbiota can have an impact on HCC tumor development, this study provides evidences showing that the tumor can also have an impact on the gut microbiota by altering it. Studies are well performed and the manuscript is well written.

Major comments:

1) The authors are showing in figure 3B the representative tumor morphology, tumor volume and liver weight. Can the authors add H&E stained sections of liver from ABX and control mice to Fig.3B?

Response: We thank for the reviewer's suggestion. We have updated the manuscript to include the H&E staining results of the liver from both the ABX-treated and control mice.

Page 8, Line 162-170

To test this hypothesis, mice were fed with water or ABX for 4 weeks to deplete gut microbiota, followed by orthotopic tumor injection. Two weeks later, mice were sacrificed, liver was removed for tumor measurement, intestine was cut for length measurement, hippocampus was isolated and subjected for LTP and STC analysis (Fig. 3A). The results showed that ABX treatment significantly reduced the HCC progression, as evidenced by decreased tumor size and total liver weight. H&E staining of the tumor section revealed that the liver cells exhibited noticeable degeneration and were loosely arranged. Focal infiltration and scattered presence of inflammatory cells within the lobules, as well as the blood vessel congestion was observed, indicating of an inflammatory microenvironment (Fig. 3B).

2) In figure 5, the authors are showing that the depletion of the microbiota in tumor-bearing mice lowers the amount of IL-1 β in the serum and in the hippocampus. However, there is no evidence showing that the level of IL-1 β in the serum of tumor-bearing mice is higher than in the control group. Have the authors measured the amount of IL-1 β in the serum of both groups (figure 1)?

Response: We thank for the reviewer's question. We agree with the reviewer that it is critical to study the expression level of IL-1 β in healthy control and tumor-bearing mice. To address this question, we collected serum samples from mice two weeks after receiving either PBS or Hepa1-6 intrahepatic injection and examined IL-1 β levels. The results showed that IL-1 β

level was significantly increased in the tumor-bearing mice, suggesting a possible association between IL-1 β levels, inflammation, and HCC.

Page 10, Line 210-215

Furthermore, our study detected elevated serum IL-1 β levels in HCC tumor-bearing mice compared to those in wild-type mice (Fig. S1), which was in agreement with a prior study showing that HCC patients had increased levels of IL-1 β compared to healthy individuals. These observations highlight the potential role of IL-1 β in the development and progression of HCC and support further investigation into the underlying mechanisms driving this association.

3) In Figure 5, the authors are showing that intestine LP leukocytes incubated with feces from tumor-bearing mice are producing more IL-1 β than when incubated with a healthy microbiota.

The amount of IL-1 β produced by Intestine LP leukocytes in vivo should also be measured. This could be assessed by isolating the intestine LP from mice harboring tumors (vs. the control group) to determine by flow cytometry the amount of IL-1 β produced by each cell population (CD11B⁺CD11C⁻, CD11B⁻CD11C⁺, CD11B⁺CD11C⁺).

Response: We appreciate the constructive suggestions from the reviewer. We encountered a number of technical difficulties during our attempts to perform flow cytometry for measuring IL-1 β production. One possible reason was the low expression levels of IL-1 β *in vivo*, making it difficult to be detected by flow cytometry. Another possible reason could be the lack of IL-1 β -specific antibody that can be applied for flow cytometry. To overcome these challenges, we employed an alternative method by measuring the levels of IL-1 β in the sorted cell populations using ELISA. Specifically, we sorted the three subpopulations from control and tumor-bearing mice, and stimulated them with LPS for 3 hours. The presence of IL-1 β in the supernatant was determined by ELISA. We found that tumor-bearing mice-derived CD11B⁺ DCs and monocytes secreted higher level of IL-1 β compared with those from control mice. Whereas CD11B⁻ DCs showed comparable levels of IL-1 β from both control and tumor-bearing mice. These results suggested that tumor-induced CD11B⁺ DCs and monocytes are in a hypersensitive stage, which was likely due to the altered microbiome. These results were also consistent with our finding that tumor-bearing mice had higher level of IL-1 β in the serum compared with control mice. Overall, while we were unable to perform flow cytometry to detect IL-1 β *in vivo*, we hope that the results we have presented in our study provided sufficient evidence that the LP leukocytes are involved in the tumor-induced IL-1 β elevation.

Page 11, Line 227-232

In addition, we also found that tumor-bearing mice-derived CD11B⁺ DCs and monocytes secreted higher level of IL-1 β compared with those from control mice after being stimulated with LPS, whereas CD11B⁻ DCs showed comparable levels of IL-1 β from both control and tumor-bearing mice (Fig. 5P). These results suggested that tumor-induced CD11B⁺ DCs and monocytes are in a hypersensitive stage, which was likely due to the altered microbiome.

4) In figure 5, the authors are showing a significant reduction of IL-1 β in the serum and in the hippocampus of the ABX group mice compare to the control group. Does the IL-1 β amount present in the hippocampus have also been assessed by IHC in both groups?

Response: We thank for the reviewer's suggestion. We have collected hippocampus samples from both Control and ABX group mice, the presence of IL-1 β in the hippocampus was confirmed by immunohistochemistry (IHC).

Page 10-11, Line 217-219

Immunohistochemical analysis also revealed a significant downregulation of IL-1 β in the hippocampal tissue of mice in the ABX group compared to the control group (Fig. 5N).

5) The authors should discuss more about by which mechanism HCC tumor can modulate the gut microbiota population.

Response: We appreciate the reviewer's comment. We agree with the reviewer regarding the importance of understanding the mechanism by which HCC tumors can modulate the gut microbiota population. To address this issue, we performed additional experiments and found that the presence of bile acids was significantly elevated in the serum and liver of the tumor-bearing mice, suggesting bile acids might mediate the alteration of microbiota population. We have incorporated these findings into the Results section and further discussed in the Discussion section.

Page 11, Line 233-240

Studies have demonstrated the ability of bile acids to promote immune responses during microbiota dysbiosis in disease development. In the present study, we aimed to investigate whether bile acids are associated with increased levels of IL-1 β in the HCC model and its potential involvement in the microbiota alteration. The results revealed that the levels of total bile acids were significantly elevated in the serum and liver of HCC tumor-bearing mice compared to control mice (Fig. 5Q), suggesting that bile acids could potentially act as pro-inflammatory factors that contribute to the increased levels of IL-1 β during the progression of HCC.

Page 16, Line 333-340

HCC often occurs in the context of chronic inflammation and activates both innate and adaptive immune responses. Studies have suggested that local and systemic immune response induced by hepatocarcinogenesis could disrupt the gut microbiome homeostasis. Moreover, we observed an increase in bile acids in the tumor-bearing mice, which played critical roles modulating microbiota composition. Bile acids could directly damage the microbiome membrane and alter the microbiota homeostasis through FXR-induced antimicrobial peptides indirectly. Further studies are required to investigate the detailed mechanism of how HCC growth affected the microbiota composition in tumor-bearing mice.

Minor Comments:

1) The number of mice used for each experiment should be indicated in each figure.

Response: We thank for the reviewer's suggestion. We have indicated the number of mice used for each experiment in the figure legends.

2) There is no mention of the Figure 3A in the text.

Response: We are sorry for the mistake and thank the reviewer for pointing it out. We have now included a description of Figure 3A in the manuscript.

Page 8, Line 162-166

To test this hypothesis, mice were fed with water or ABX for 4 weeks to deplete gut microbiota, followed by orthotopical tumor injection. Two weeks later, mice were sacrificed, liver was removed for tumor measurement, intestine was cut for length measurement, hippocampus was isolated and subjected for LTP and STC analysis (Fig. 3A).

3) For the Figure 5N, the name of each immune cell type should be mentioned in the figure or in the legend, and the gating strategy used to isolate each cell population should be provided as supplementary information.

Response: We thank for the reviewer's suggestions. We have named the CD11B⁺CD11C⁻ cells as monocytes, CD11B⁺CD11C⁺ cells as CD11B⁺ DCs, CD11B⁻CD11C⁺ cells as CD11B⁻ DCs. Furthermore, we have included a detailed flow gating strategy in Figure S2 to help clarify the methodology that used to differentiated these three populations.

Page 11, Line 222-224

we sorted the LP CD11B⁺CD11C⁻ monocytes, CD11B⁻CD11C⁺ dendritic cells, and CD11B⁺CD11C⁺ dendritic cells from WT mice and stimulated with fecal contents isolated from mock or tumor-bearing mice, respectively (Fig. S2).

Reviewer #2:

In this study, Mei and colleagues used a preclinical model to evaluate the role of the gut microbiota in mediating the cognitive disruption that is commonly observed in patients with neoplastic lesions. They used a mouse model of hepatocellular carcinoma to evaluate the contribution of the gut microbiota to changes in the electrophysiology recordings as surrogate of synaptic plasticity. Using, antibiotic-mediated depletion of the gut microbiota as well as fecal transplants in antibiotic-treated mice, authors elegantly demonstrated the involvement of the gut microbiota in the observed changes in synaptic tagging/capture recordings of mice with induced hepatocellular carcinoma. Authors also demonstrated that these physiological alterations were likely mediated by the microbiota inducing IL1b production in the gut of animals with hepatocellular carcinoma. Overall, the paper is well-written, the methods used are sound and the research rigorous with no overstatement. I have a few suggestions for the authors:

1) IL1b has been shown to impact blood brain barrier, however this is not considered by the authors in the discussion. I believe this is an important point to address as blood brain barrier leakage would facilitate the translocation of toxic byproducts that can further exacerbate the observed phenotype.

Response: we appreciate the reviewer's insightful feedback and agree that it is important to consider the impact of IL-1 β on the BBB and the transport of toxic byproducts into the CNS. We have now discussed this perspective in the Discussion section.

Page 14, Line 288-296

IL-1 β played a critical role in BBB dysfunction. IL-1 β could activate the cell adhesion molecules such as intercellular adhesion molecule 1 (ICAM-1) and vascular cell adhesion molecule 1 (VCAM-1) in the astrocytes, leading to an increased leukocyte adhesion to the BBB endothelium and their subsequent infiltration into the CNS. It could also promote the secretion of other pro-inflammatory cytokines such as IL-6 and TNF- α to disrupt the paracellular BBB pathway, increasing the BBB permeability that could allow the harmful toxins to enter the CNS. In this current study, we found that HCC tumor-bearing mice had higher serum levels of IL-1 β , which could disrupt BBB and allow the bile acids to infiltrate the brain. Whether IL-1 β and bile acids could synergistically disrupt BBB permeability warrants further investigation.

2) Just wondering whether the authors have considered bile acid metabolism as mediator of both the initial changes in the gut microbiota of mice with hepatocellular carcinoma as well as the observed cognitive phenotype. Altered bile acid metabolism is associated with altered liver function. High levels of bile acids in the gut are considered proinflammatory and have known antimicrobial effects. On the other hand, some antibiotics are known to induce cholestasis, which would retain bile acids in the liver.

Response: We thank the reviewer for the insightful comments on the possible involvement of bile acids in the initial changes in the gut microbiota. In response to this suggestion, we measured the levels of bile acids in the serum and liver. We found that tumor-bearing mice had significantly higher total bile acid levels in both serum and liver compared to control

mice. These results suggested that changes in bile acid metabolism may play a role in the observed alterations in gut microbiota and cognitive impairment in mice with HCC tumors. We have added these results in our manuscript and discussed this phenomenon in the Discussion section.

Page 11, Line 233-240

Studies have demonstrated the ability of bile acids to promote immune responses during microbiota dysbiosis in disease development. In the present study, we aimed to investigate whether bile acids are associated with increased levels of IL-1 β in the HCC model and its potential involvement in the microbiota alteration. The results revealed that the levels of total bile acids were significantly elevated in the serum and liver of HCC tumor-bearing mice compared to control mice (Fig. 5Q), suggesting that bile acids could potentially act as pro-inflammatory factors that contribute to the increased levels of IL-1 β during the progression of HCC.

Page 13-14, Line 272-287

In addition to ammonia, other factors such as short-chain fatty acids, manganese and bile acids have also been reported to contribute to the HE development. Among these factors, bile acids have gained significant scientific interest in recent years. Bile acids are a group of compounds synthesized in the liver and secreted into the digestive tract to aid in the digestion and absorption of dietary fats, vitamins and xenobiotics. Studies have shown that bile acids can disrupt BBB (blood brain barrier) integrity and subsequently expose neurons to potentially toxic molecules. Additionally, bile acids acted as peripheral signaling molecules that activate bile acid receptors present in both BBB and brain, further impacting BBB permeability and brain function. In this current study, we observed an increase in total bile acid levels in the liver and peripheral blood of HCC tumor-bearing mice compared to control mice, indicating that HCC tumor growth affected bile acid secretion and distribution. This finding is consistent with a previous report that increased bile acid promoted IL-1 β secretion in DEN-induced murine HCC model. On the other hand, some studies have shown that certain types of bile acids exhibited anti-inflammatory properties by activating FXR and GPBAR1 signaling pathways. Further investigation is needed to identify the specific bile acid components present in HCC tumor-bearing mice that may contribute to hippocampal cognitive impairment.

3) In figure 2A-B, please add the profiles of all the animals rather than the averaged abundance. Also Figure 2D and 2E are redundant as they show the same effect using different methods. Please keep just one.

Response: We thank for the reviewer's suggestions. We have updated Figure 2A-C to include the microbiota composition of each individual mouse. Additionally, we have decided to remove the NMDS plot from the manuscript. These changes should help to better highlight the key findings of our study.

4) Figure 5A-L. Are differences for any inflammatory marker significant? IL6 and IL17 behave in the same way as IL1b, but they are not further considered by the authors. Thoughts?

Response: We appreciate the reviewer's insightful comments on the potential effects of IL-6 and IL-17 in our study that we did not analyze in the previous manuscript. We agree that these cytokines displayed a similar trend of expression as IL-1 β , but the statistical analysis revealed that the p-values were 0.17 (IL-6) and 0.10 (IL-17), respectively. We have also tried to remove the outliers from the analysis, but the difference remained statistical insignificant. Nevertheless, we agree with the reviewer that the involvement of these cytokines cannot be fully excluded. Therefore, we have described this in the Result section and discussed the potential role of IL-6 and IL-17 in the Discussion section of the manuscript.

Page 10, Line 206-210

Significant reduction of the pro-inflammatory cytokine IL-1 β was observed in the serum of the mice in the ABX group. While other pro-inflammatory cytokines such as IL-6 and IL-17 showed a decreasing trend, the differences were not statistically significant. No differences were observed in the levels of the remaining cytokines, including IL-1 α , IL-10, IL-12, IL-23, IL-27, IFN- β , IFN- γ , TNF- α , and GM-CSF (Fig. 5A-L).

Page 19-20, Line 410-418

In addition to IL-1 β , a decreasing trend of the pro-inflammatory cytokines IL-6 and IL-17 was also observed in the ABX group mice. These cytokines have been implicated in the gut-liver axis during liver pathogenesis. For example, altered gut microbiota-derived short-chain fatty acid have been shown to suppress IL-17 expression by $\gamma\delta$ T cells, which are critical in the development of HCC. Tans-signalling of IL-6 has also been found to be essential for HCC progression. Notably, single depletion of IL-1 β with the neutralization antibody could sufficiently restore the STC in the tumor-bearing mice, further ABX treatment failed to rescue the STC to a higher extend, suggesting that IL-1 β is the dominant pro-inflammatory cytokine mediating STC decline in the tumor-bearing mice.

5) Please carefully check the manuscript for typos etc. E.g. Line 217 tumor-nearing should be tumor-bearing(?)

Response: We thank the reviewer for the comments. We apologize for the typo in the previous version of our manuscript. We have carefully revised the English writing and corrected the typos throughout the manuscript.

6) Sequencing data should be deposited in a public repositorie (e.g. SRA).

Response: We thank for the reviewer's suggestion. We are fully aware that data sharing is an important aspect of scientific research and have deposited our sequencing data in the BIG database under the project number: PRJCA015135.

Reviewer #3:

The paper by Mei and Colleagues is a study on a very hot topic, related to the effects of gut microbiota in the brain, in the presence of a tumor-induced state of inflammation.

The study is well organized and different experimental approaches are used to verify the principal hypothesis of a role for IL-1b as mediator of the gut-brain communication in a model of hepatocellular carcinoma.

However, many weak points need to be addressed before considering the manuscript suitable for the publication.

Major points:

1) English must be deeply revised

Response: We thank for the reviewer's suggestion. To improve the language quality, we have made extensive and careful revisions to the manuscript. Additionally, we have engaged the assistance of two native English-speaking researchers to check our writing for clarity, accuracy, and appropriateness of scientific terminology. We hope that these efforts have led to a significant improvement of our manuscript.

2) More than one cell line should be used in the animal models.

Response: We thank for the reviewer's suggestions. We agree with the reviewer that utilizing an alternative HCC model could provide additional support for our findings. Thus, we attempted to establish another murine HCC model using the Hepa1c1c7 cell line obtained from ATCC. Unfortunately, our efforts were unsuccessful as we were unable to achieve orthotopic growth of the cells in the liver, even after increasing the numbers of cells injected. Hence, we discussed this as one of our study limitations in the Discussion section.

Page 15, Line 318-323

In this study, we established a murine orthotopic HCC model by directly injecting Hepa1-6 cells into the liver to investigate the influence of HCC growth on gut microbiota composition and diversity. This orthotopic HCC model is superior to the widely used subcutaneous model as it more accurately recapitulates the complex microenvironmental features of the tumor, including the vasculature, stroma, and surrounding lymphatic system, etc.

Page 16, Line 342-347

However, it is important to note that the extent and specificity of these changes may vary depending on the particular tumor model being investigated. Additional studies are needed to determine whether these findings can be generalized across different HCC models. Nevertheless, the similarities overserved in the present study and in other HCC models suggested that these results may have broader implications and warrant further investigations.

3) Too low number of mice is used for the microbiote analyses. Acceptable numbers range from 10 to 20.

Response: We thank the reviewer for the valuable suggestion. We agree with the reviewer that the sample size of mice used in our study was insufficient for robust data analysis. To address this concern, we conducted additional 16S rDNA sequencing experiments with a larger sample size of 10 mice in each group. Our results, which are in agreement with the previous results, are presented in Figure 2. These updated data provide a more comprehensive understanding of the alterations in microbiota during HCC development and strengthen the conclusions in our manuscript.

Page 7-8, Line 132-158

The taxonomy profiles at the class level revealed that the intestine bacteria community was dominated by Bacteroidia (55.5% on average), followed by Clostridia (24.8% on average), Bacilli (15.4% on average) and Actinobacteria (2.7% on average); at the genus level, the intestine bacteria community was dominated by Muribaculaceae (45.5% on average), Lactobacillus (8.6% on average), Alistipes (5.7% on average), Lachnospiraceae_NK4A136_group (5.2% on average), followed by f_Lachnospiraceae_Unclassified (4.4% on average) and Dubosiella (4.3% on average). During the HCC progression, we observed significant changes in the relative abundance of several bacterial taxa. Specifically, the relative abundance of Bacilli (2.4 folds on average) and Verrucomicrobiae (1.7 folds on average) decreased most dramatically at class level, [Eubacterium]_fissicatena_group (9.24 folds on average), Dubosiella (4.1 folds on average) and Lactobacillus (2.35 folds on average) at genus level. Conversely, the relative abundance of Bacteroides (617.1 folds on average) was most dramatically increased at genus level (Fig. 2A).

Next, we used different index to examine the alpha diversity of the microbiota in control and tumor-bearing mice. As shown in Fig. 2B, the ACE index, Shannon index and Simpson index showed no significant difference within control and tumor-bearing mice. Interestingly, we noticed that the observed species were significantly decreased in the tumor-bearing mice, suggesting the microbiota diversity was decreased during cancer progression. We also examined the taxonomic beta diversity between the groups. A significant difference was observed by weighted UniFract analysis (Fig. 2C). PCA analysis also demonstrated that the microbiota community of control and tumor-bearing mice showed clear separation (Fig. 2D). Using linear discriminant analysis (LDA) analysis, we identified differential bacteria taxa between control and tumor-bearing mice. Specifically, we found that c_Bacilli and o_Lactobacillales were enriched in the fecal samples from control mice, while o_Oscillospiraceae and f_Bacteroidaceae were enriched in the fecal samples from tumor-bearing mice (Fig. 2E). Collectively, our results demonstrated that the microbiota diversity and composition were altered during HCC cancer progression.

4) *Weak statistical analyses. The authors never mentioned the number of samples/repeats performed.*

Response: We appreciate the reviewer's comments regarding the statistical analyses in our paper. We apologize for not providing sufficient information for the number of samples/repeats in our original manuscript. To address this concern, we have revised our graphical representation to show individual data points using a dot plot (Fig. 3B&C, Fig. 5Q). We have also updated the figure legends to include the number of samples/repeats for each experimental group. We believe that these revisions provide a more accurate representation of the data and help to avoid any potential misunderstandings.

5) *The only cytokine modified by Abx treatment is IL1b, as shown in fig 5. Is the difference in IL1b still significant if the outlier in control is eliminated?*

Response: We understand the reviewer's concern that the statistical significance of the IL-1 β level may have been influenced by the outlier in the Ctrl group. To address this concern, we re-analyzed the data by omitting the outlier. The average IL-1 β level in the Ctrl group was 12.76 ± 6.58 pg/mL, while the average IL-1 β level in the Abx group was 1.58 ± 2.17 pg/mL, with a p-value of 0.018. This re-analysis confirmed that the antibiotic treatment significantly reduced the IL-1 β levels compared to the control group even when the outlier was excluded.

REVIEWERS' COMMENTS:

Reviewer #1 (Remarks to the Author):

The authors have addressed all my comments.

Reviewer #2 (Remarks to the Author):

The authors have reasonably addressed all the comments I raised in the first round of review. No further comments from my side.

Reviewer #3 (Remarks to the Author):

In this revised version, the authors answered all my concerns. I am satisfied with the revision of the new version of the manuscript.

Dear Editor,

We would like to express our gratitude to the reviewers for their valuable feedback and comments on our manuscript. We are pleased to see that all three reviewers have acknowledged our efforts and have indicated that we have adequately addressed their concerns. We would like to thank each reviewer for their time and attention to detail in reviewing our work.

Reviewer #1 (Remarks to the Author):

The authors have addressed all my comments.

Response: We appreciate your acknowledgment that we have addressed all of your comments.

Reviewer #2 (Remarks to the Author):

The authors have reasonably addressed all the comments I raised in the first round of review. No further comments from my side.

Response: Thank you for noting that we have reasonably addressed all the comments from the previous review.

Reviewer #3 (Remarks to the Author):

In this revised version, the authors answered all my concerns. I am satisfied with the revision of the new version of the manuscript.

Response: We are delighted to hear that you are satisfied with the revision of the new version of the manuscript.

Once again, we would like to express our sincere gratitude to the reviewers for their constructive feedback, which has undoubtedly strengthened our manuscript. We believe that the revisions made based on their suggestions have significantly improved the clarity and quality of our research. We are confident that our manuscript is now in a much stronger position for publication.

Sincerely,

Haiyan Liu